# Identification and Molecular Characterization of a Novel Large-Scale Variant (Exons 4_18 Loss) in the LDLR Gene as a Cause of Familial Hypercholesterolaemia in an Italian Family

**DOI:** 10.3390/genes14061275

**Published:** 2023-06-16

**Authors:** Paola Concolino, Elisa De Paolis, Simona Moffa, Maria Elisabetta Onori, Laura Soldovieri, Claudio Ricciardi Tenore, Maria De Bonis, Claudio Rabacchi, Concetta Santonocito, Martina Rinelli, Sebastiano Calandra, Andrea Giaccari, Andrea Urbani, Angelo Minucci

**Affiliations:** 1Departmental Unit of Molecular and Genomic Diagnostics, Fondazione Policlinico Universitario A. Gemelli IRCCS, 00168 Rome, Italy; paola.concolino@policlinicogemelli.it (P.C.); elisa.depaolis@policlinicogemelli.it (E.D.P.); mariaelisabetta.onori@guest.policlinicogemelli.it (M.E.O.); claudio.ricciarditenore@guest.policlinicogemelli.it (C.R.T.); maria.debonis@policlinicogemelli.it (M.D.B.); concetta.santonocito@policlinicogemelli.it (C.S.); martina.rinelli@guest.policlinicogemelli.it (M.R.); 2Genomics Core Facility, Gemelli Science and Technology Park (G-STeP), Fondazione Policlinico Universitario A. Gemelli IRCCS, 00168 Rome, Italy; 3Center for Endocrine and Metabolic Diseases, Fondazione Policlinico Universitario A. Gemelli IRCCS, 00168 Rome, Italy; simona.moffa@guest.policlinicogemelli.it (S.M.); andrea.giaccari@policlinicogemelli.it (A.G.); 4Department of Medical and Surgical Sciences for Children and Adults, University of Modena and Reggio Emilia, 41121 Modena, Italy; claudio.rabacchi@unimore.it; 5Department of Basic Biotechnological Sciences, Intensivological and Perioperative Clinics, Università Cattolica del Sacro Cuore, 00168 Roma, Italy; 6Department of Biomedical, Metabolic and Neural Sciences, University of Modena and Reggio Emilia, 41121 Modena, Italy; sebcal@unimore.it

**Keywords:** familial hypercholesterolemia, copy number variations (CNVs), next-generation sequencing, *LDLR* gene, LDL cholesterol, Alu sequences

## Abstract

Next-generation sequencing (NGS) is nowadays commonly used for clinical purposes, and represents an efficient approach for the molecular diagnosis of familial hypercholesterolemia (FH). Although the dominant form of the disease is mostly due to the low-density lipoprotein receptor (LDLR) small-scale pathogenic variants, the copy number variations (CNVs) represent the underlying molecular defects in approximately 10% of FH cases. Here, we reported a novel large deletion in the LDLR gene involving exons 4–18, identified by the bioinformatic analysis of NGS data in an Italian family. A long PCR strategy was employed for the breakpoint region analysis where an insertion of six nucleotides (TTCACT) was found. Two Alu sequences, identified within intron 3 and exon 18, could underlie the identified rearrangement by a nonallelic homologous recombination (NAHR) mechanism. NGS proved to be an effective tool suitable for the identification of CNVs, together with small-scale alterations in the FH-related genes. For this purpose, the use and implementation of this cost-effective, efficient molecular approach meets the clinical need for personalized diagnosis in FH cases.

## 1. Introduction

Familial hypercholesterolemia (FH) is a dominant hereditary metabolic disorder usually related to pathogenic variants in the *LDLR* (low-density lipoprotein receptor), *APOB* (apolipoprotein B) or *PCSK9* (proprotein convertase subtilisin/kexin type 9) genes. The prevalence was initially inferred to be 1:500 for heterozygous FH (HeFH) and 1:1,000,000 for homozygous FH (HoFH). While HoFH is still a rare condition with a prevalence of 1:160–400,000, the current HeFH prevalence is 1:313 [1,2].

The well-known typical clinical features of FH are high levels of total and LDL cholesterol. For ASCVD (atherosclerotic cardiovascular disease) prevention, the early detection and the timely beginning of a lipid-lowering treatment is essential, with the final aim being the reduction of the LDL cholesterol level below 55 mg/dL (1.42 mmol/L) or below 70 mg/dL (1.81 mmol/L), according to the individual risk [3]. The Dutch Lipid Clinical Network Criteria (DLCN), the Simon Broome Criteria (SB), and the Making Early Diagnosis Prevents Early Death (MEDPED) are the universally accepted diagnostic algorithms. Genetic testing represents a key step of all these clinical diagnoses [4]. Additionally, the Familial Hypercholesterolemia Foundation has recommended the use of genetic testing as the standard of management in subjects with a FH definitive or a suggestive diagnosis, and for their relatives [5,6].

FH is mainly related to the occurrence of loss-of-function alterations in the *LDLR* (60–80% of patients) and *APOB* (5–10%) genes or gain-of-function variants in the *PCSK9* gene (<1%). Rarely, other genes such as the *APOE*, *LDLRAP1*, *LIPA* and *SCAP* are involved [7]. The prevalent types of mutation in the FH-related genes are missense, indels and splicing. Copy number variations (CNVs) are the underlying cause of the FH condition in approximately 10% of cases. CNVs exist in genomic structural variants as deletions and duplications with a size larger than 50 bp. The role of CNVs in dyslipidemias is still being explored. In 2018, Iacocca et al. reported several unique deletions and duplications in the *LDLR* gene [8]. This finding was probably related to the presence in the LDLR gene of 98 Alu repeats: 95 intronic and 3 located in the 3′ untranslated region (UTR) (9). The *LDLR* gene is particularly prone to CNVs and the breakpoints sites generally occur in the intronic regions, resulting in whole-exon deletions [9]. MLPA (multiplex ligation-dependent probe amplification) or aCGH (array-based comparative genome hybridization) are considered the gold standard methods for the identification of CNVs. However, the use of bioinformatics prediction of CNV events from next-generation sequencing (NGS) data is now also a common laboratory practice.

In this paper, we studied an Italian patient with a personal and familial history of hypercholesterolemia, a carrier of a novel large-scale variant (exons 4_18 loss) in the *LDLR* gene.

## 2. Materials and Methods

### 2.1. Case Presentation and Family History

The proband was a 34-year-old Italian woman referred to our Center for Endocrine and Metabolic Diseases in November 2022. When she was 6 years old (in 1994), a routine blood test revealed a hypercholesterolemic status indicated by elevated plasma lipid values corresponding to total cholesterol (TC) 374 mg/dL, high density lipoprotein (HDL) 93 mg/dL, triglyceride (TGs) 281 mg/dL and, low density lipoprotein (LDL-C) 225 mg/dL as shown in Table 1.

The proband had not been treated until the age of 18 years, when she started treatment with a daily 10 mg dose of Ezetimibe. When she was 20 years old, a low dose of statin was added to Ezetimibe by her endocrinologist. The treatment was soon stopped as the consequence was intolerable muscle pain, although this was not correlated with a CPK elevation. Since then, she has been treated with Ezetimibe in monotherapy. In 2019, her lipid profile remained unchanged, with very high levels of TC and LDL-C, as shown in Table 1.

In 2021, the proband started treatment with the anti-PCSK-9 monoclonal antibody Alirocumab 75 mg every two weeks, which was then increased to 150 mg every two weeks; this is still ongoing.

As reported in Table 1, despite the treatment with Alirocumab, the proband’s lipid profile is not yet at the target levels, according to the European Society of Cardiology (ESC)/ European Atherosclerosis Society (EAS) guidelines of 2016 and 2019 [3]; therefore, we decided to again try the low-dose statin therapy adding Rosuvastatin 5 mg daily to Alirocumab and Ezetimibe. This was a new attempt after the experienced muscle pain during the simvastatin, atorvastatin, and lovastatin therapy. We decided to treat with a statin with a different metabolism, which is still ongoing. In case the patient would not tolerate even Rosuvastatin, we planned to give her Bempedoic Acid, a novel hypolipidemic drug (recently made reimbursable by the Italian national health care system) that inhibits the enzymatic activity of ATP citrate lyase in the cholesterol synthesis pathway, which is not activated in the muscle. Therefore, we thought that the patient could tolerate it, although this drug has a reduction effect on LDL cholesterol close to 20 percent, waiting for more effective drugs such as Evinacumab [10,11].

On physical examination, no tendon xanthoma or corneal arcus were observed. The proband had a body mass index of 30 kg/m^2^ and no history of coronary heart disease; she was normotensive, non-diabetic, and did not smoke. Diagnosis of FH was made considering a lipid profile strongly suggestive for FH and a score of 12 [12]. In line with the EAS/European Society of Cardiology (ESC) guidelines, the patient was addressed to genetic evaluation. The above-mentioned guidelines recommend molecular evaluation in the presence of: TC > 8 mmol/L (>310 mg/dL) without treatment in an adult or adult family member (or >95th percentile by age and gender for country); premature CHD in the patient or a family member; tendon xanthomas in the patient or a family member; or sudden premature cardiac death in a family member [3].

The family pedigree is reported in Figure 1. The proband’s paternal great-grandmother was the first family member known to suffer from hypercholesterolemia. She had five children, four sons and one daughter. Three of them died from acute myocardial infarction between the age of 37 and 50 years; among these was the proband’s grandfather who died at the age of 46 years.

The proband inherited the hypercholesterolemia from her father who is currently treated with Rosuvastatin 10 mg and Ezetimibe 10 mg daily. He has two siblings: one brother who dead at the age of 46 years from myocardial infarction and one sister, who is still alive and treated with statins. Written informed consent was obtained before genetic testing.

### 2.2. NGS and Large Genomic Rearrangement (LGR) Detection

Genomic DNA was isolated from peripheral blood using MagCore^®^ Genomic DNA whole blood kit (RBC Bioscience, New Taipei City, Taiwan) on the automated MagCore^®^ HF16Plus instrument (Diatech Lab Line, Jesi, Italy) following the manufacturer’s instructions. The quantitation of the extracted DNA was performed using the Qubit dsDNA BR fluorimetric assays (Life Technologies, Gaithersburg, MD, USA).

The multi-genes panel Devyser FH NGS kit v2 (Devyser, Hagersten, Sweden) was adopted for the targeted NGS (tNGS) analysis. The multi-gene panel covered the coding regions and splicing junctions of the following genes: *LDLR* (NM_000527.4), *APOB* (NM_000384.2), *PCSK9* (NM_174936.3), *APOE* (NM_000041.3), *LDLRAP1* (NM_015627.2), and *STAP1* (NM_012108.3) and the sequence determination of the following polygenic single nucleotide polymorphisms (SNPs) related to FH and/or statin treatment response: rs629301, rs1564348, rs1800562, rs2479409, rs3757354, rs4299376, rs6511720, rs8017377, rs11220462, rs1367117, rs429358, rs7412, rs646776, rs4149056, rs3798220, rs10455872. NGS was performed on the Illumina MiSeq^®^ NGS platform (Illumina, San Diego, CA, USA). FASTQ files were analysed by the CE-IVD Amplicon Suite Software (SmartSeq, Novara, Italy), as previously reported [13].

The MLPA assay was performed as a confirmatory method of the new rearrangement using the SALSA MLPA kit for *LDLR* (P062; MRC Holland, Amsterdam, The Netherlands) on an ABI 3500 Genetic Analyzer (Thermo Fisher Scientific, Foster City, CA, USA). The collected data were evaluated using Coffalyser.NET Software (MRC Holland). Three healthy males and three healthy females were included as wild-type controls.

### 2.3. Analysis of Breakpoint Region

The characterization of the breakpoint region was performed using deletion-specific primers (DF 5′–ACCGCTGCATTCCTCAGTTCTG–3′ and DR 5′–AACCTGAAGTCCCGTCAAAC–3′) using a long-range PCR protocol (Expand Long Template PCR System, Roche Applied Science, Monza, Italy). Sequencing was performed using a BigDye Terminator Cycle Sequencing Kit v3.1 (Thermo Fisher Scientific) on the ABI 3500 Genetic Analyzer (Thermo Fisher Scientific). Internal primer sequence is available on request. Results were analyzed with the SeqScape v2.5 software package (Thermo Fisher Scientific) using NG_009060.1 as reference. The Repeat Masker program was employed to identify *Alu* sequences at breakpoint junctions [14].

## 3. Results

### 3.1. NGS Analysis and LRG Detection

No small-scale pathogenic variants were detected in the six FH-related genes investigated by the Devyser FH NGS kit. From the NGS CNV analysis emerged the presence of a large LDLR deletion, involving exons 4–18 (Figure 2a). This result was confirmed by performing the MLPA assay on a fresh DNA sample (Figure 2b). Successively, the proband’s father also resulted as a carrier of the deletion at the LDLR MLPA analysis.

### 3.2. Analysis of Breakpoint Region

A PCR fragment of 891 bp showed a wild-type sequence until the nucleotide g.19163T (NG_009060.1) of *LDLR* intron 3. The following sequence showed an upstream insertion of 6 nucleotides (TTCACT) to the sequence corresponding to the *LDLR* exon 18 region starting from the g.48821A (NG_009060.1) nucleotide (Figure 3a). According to the HGVS nomenclature, we reported the rearrangement as NG_009060.1:g.19164_48820delinsTTCACT. The Repeat Masker program identified two *Alu* sequences around breakpoint junctions: an *AluSx* within intron 3 and an *AluJb* in exon 18 of the *LDLR* gene. Figure 3b shows the sequence homology near the breakpoint site of *LDLR* 4_18del.

## 4. Discussion

In this study, the molecular workflow allowing the diagnosis of a patient with a personal and family history of FH was reported. A NGS-based approach with appropriate bioinformatics analysis detected a previously unreported large-scale deletion in the *LDLR* gene. The structural rearrangement identified involves a 29.6-kb region, causing the loss of the exon 4–exon 18 gene regions. In addition, by a long-PCR strategy, we also established the exact genomic coordinates of the deletion characterized by a six-nucleotide insertion (TTCACT) at the breakpoint site: NG_009060.1:g.19164_48820delinsTTCACT (Figure 3a). Exons 4–18 represent a large part of the LDLR coded protein, which are predicted to involve the relevant domains as part of the ligand-binding region (exons 4–6), the EGF precursor homology region (exons 7–14), the O-linked glycan region (exon 15), the transmembrane domain (exon 16), and the cytosolic domain (exons 17–18). Since most of the functional domains of the protein were lost, this rearrangement was considered pathogenic, supporting the molecular diagnosis of HeFH in our patient [15,16].

The *LDLR* gene lies on human chromosome 19 (19p13.1-3), which is the one presenting the highest percentage of *Alu* repeats [17,18]. *Alu* repeats account for 65% of the *LDLR* intronic region, reaching 85% of the genomic sequence outside the exon–intron junctions [9]. Shortly after the *LDLR* gene cloning, the first Southern blotting assays defined that *Alu* repeats mediate unequal meiotic cross-over and lead to large gene rearrangements, causing FH [19]. Basically, these sequences offer many opportunities for homologous recombinations, and nonallelic homologous recombination (NAHR) represents a common disease-causing mechanism associated with genome rearrangements [9,20]. We identified two *Alu* sequences located around the breakpoint junctions, an *AluSx* within intron 3 and an *AluJb* in exon 18 of the *LDLR* gene, supporting the hypothesis that a homologous recombination event could underlie the identified rearrangement. However, as reported in Figure 3b, a low homology was found close to the breakpoint site. In addition, an insertion of six nucleotides (TTCACT) characterized the novel rearrangement (Figure 3).

This case report depicted that the improvements of sequencing technology resulted in a noteworthy enhancement in the turnaround time, throughput, and detection of all the possible causes of disease gene defects in FH-related genes [21]. To date, it is known that about 10% of the disease-causing variants consist of *LDLR* CNVs, so the opportunity for the simultaneous detection of the entire spectrum of the *LDLR* variants is crucial for achieving a definitive molecular diagnosis of FH. Although the MLPA assay remains the reference method for the assessment of CNVs, Iacocca et al. demonstrated that NGS data from FH patients have a 100% concordance rate for large-scale *LDLR* CNVs calling using MLPA as the “gold standard” reference method [8]. The methodological approach described in the paper is consistent with the state-of-art diagnostic workflow adopting NGS for a comprehensive analysis of SNVs and CNVs, coupled with confirmatory MLPA [22]. Here, we provided additional information about the breakpoints characterization and the possible underlying molecular mechanism. In our opinion, NGS bioinformatic analysis is nowadays a confident approach for the effective identification of large and multi-exons deletion and amplification. Likewise, the use of MLPA as an orthogonal confirmatory test is still mandatory, especially in case of single-exon or small exonic rearrangement.

We emphasize that the careful analysis of CNVs from NGS data is a relevant topic, considering that not all the sequencing facilities and the clinical laboratories have the resources and the time to set up an ad hoc parallel MLPA workflow to detect them. In particular, using NGS data to detect CNVs would delete the cost for the *LDLR* MLPA analysis, which is approximately USD 80 per sample. For this reason, it is important that all laboratories that already use the NGS technology, or that are currently in the process of doing so, also implement their workflow to the NGS-based CNV evaluation. Firstly, an internal validation of the CNV calling should be performed on previously genotyped samples, so that the sensitivity and specificity of the method can be calculated [23,24].

Furthermore, CNV evaluation can be extended to all the other FH-associated genes such as *APOB*, *PCSK9*, *LDLRAP1*, and *APOE* at no extra cost. The occurrence of CNVs in these genes are rare. However, we underlined that they have long remained uninvestigated. Extending the CNV analysis to all these FH-associated genes will increase the ability to identify all genetic aberrations that can explain FH cases [25,26].

Finally, we believe that the progressive, and hopefully inexorable, consolidation of NGS as a widespread method of choice in multiple diagnostics context, will have a strong impact on the know-how of FH, enabling the setup of public health approaches aimed primarily at the early clinical diagnosis and treatment of FH. This would benefit all populations, even those little-studied populations for which the genetics of FH is still unexplored and poorly understood.

## 5. Conclusions

This study showed that the integrated care model routinely adopted in our laboratory for the molecular diagnosis of FH led to the identification of a novel *LDLR* 29.6-kilobase deletion in an Italian family, contributing to broadening the mutational *LDLR* landscape in our population.

The NGS-based approach has the ability to effectively detect CNVs in the *LDLR* gene and could be adopted to extend the CNV screening to other FH-related genes. Overall, the molecular findings that emerged from the bioinformatic analysis still need to be confirmed by an orthogonal MLPA assay, which remains so far the reference method for CNV assessment.

## Figures and Tables

**Figure 1 genes-14-01275-f001:**
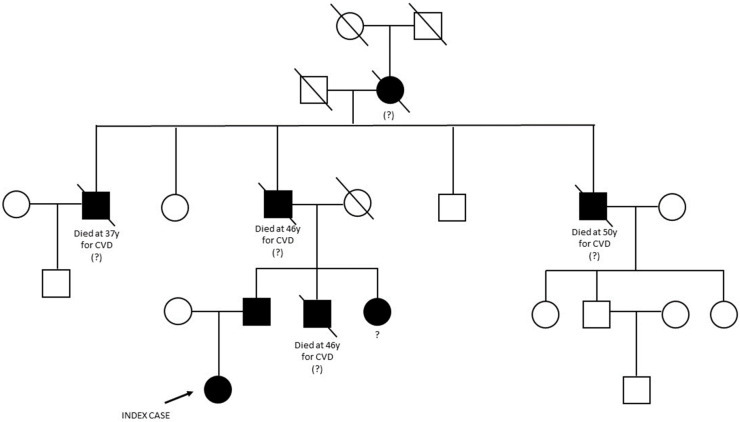
The figure illustrates the proband’s family tree. Molecular genetic diagnosis was performed only in the proband (indicated by the arrow) and her father. Several family members prematurely died from CVD (cardiovascular disease). Question marks indicate the family members with suspected FH without a molecular diagnosis.

**Figure 2 genes-14-01275-f002:**
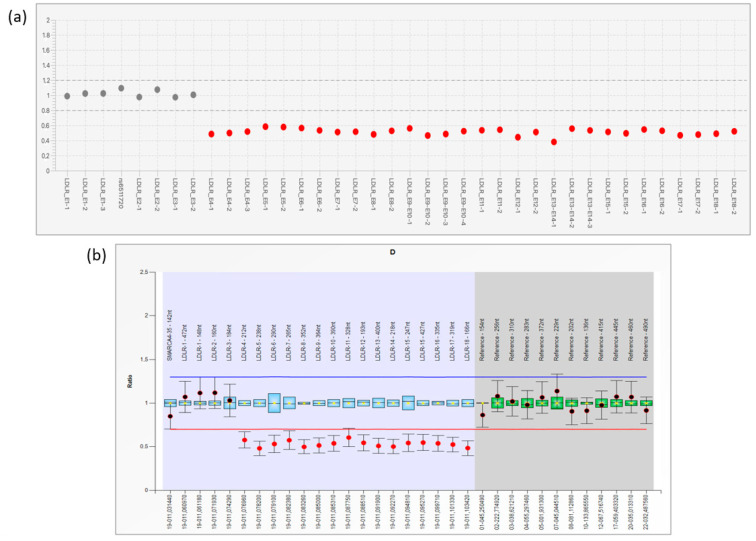
(**a**): Bioinformatics CNVs prediction of the *LDLR* gene performed by Amplicon Suite^®^ tool. Copy number plot of the *LDLR* gene with indication of heterozygous deletion of exons 4–18. (**b**): MLPA results obtained by Coffalyser.NET Software. The final ratio of 0.50 for *LDLR* exons 4–18 probes highlighted that a deletion occurred in heterozygous status (normal range 0.80 < FR < 1.20).

**Figure 3 genes-14-01275-f003:**
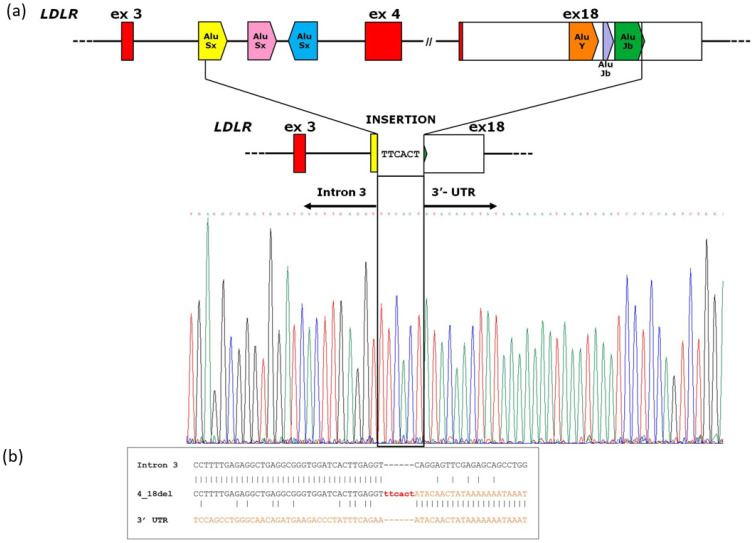
(**a**): Schematic representation of the deletion of *LDLR* exon 4_18. The panel shows: the gene region spanning from exon 3 to exon 18, as well as the location of the *Alu* sequences (arrowed box) involved in the deletion; the gene region resulting from the deletion; the electropherogram of the breakpoint: *LDLR* NG_009060.1:g.19164_48820delinsTTCACT. The red boxes represent coding sequence, white boxes 3′UTR region and lines introns. (**b**): Sequence alignment of *LDLR* Intron 3 and *LDLR* 3′–UTR regions. The figure depicts the sequence homology near the breakpoint site of *LDLR* 4_18del.

**Table 1 genes-14-01275-t001:** Lipid profiles of the proband. The first column shows the lipid profile off therapy, when she was 6 years old, the second one shows the results with Ezetimibe 10 mg therapy, and the last column indicates the lipid profile after adding Alirocumab 150 mg every two weeks.

Lipid Profile	Years			Normal Range
	1994	2019	2022	
Total Cholesterol (mg/dL)	374	357	321	130–200 mg/dL
HDL	93	68	66	female > 45; male > 40
Triglycerides (mg/dL)	281	124	95	20–170 mg/dL
LDL (mg/dL)	225	264	236	<130 mg/dL

## Data Availability

Data are available from the Authors upon request.

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
