# Peer review of "Identification and Molecular Characterization of a Novel Large-Scale Variant (Exons 4_18 Loss) in the LDLR Gene as a Cause of Familial Hypercholesterolaemia in an Italian Family"

_genes, 2023, doi:10.3390/genes14061275_

Round 1

Reviewer 1 Report

The authors reported  a novel large deletion of the LDLR gene 29 involving exons 4–18, identified by bioinformatic analysis of NGS data in an Italian family. Using the NGS to identify CNVs is very important, however, there are concerns listed below.

1. There are some papers about identify the CNVs in NGS data (such as Genes (Basel). 2022;13(8):1424), Is there anything special about the methods or techniques in your article compared to other articles? what are the highlights of your article?

2. For the proband, I'm interesting in her LDL-C levels. It seemed that PCSK9i didn't work for this patient. The first LDL-C level was 225mg/dl in 1994, but it still had 236mg/dl in 2022. Could you explain why used Ezetimibe plus Alirocumab can't reduce LDL-C levels? Might because the large deletion of the LDLR gene cause the dysfunction of LDLR? But it is a heterozygous variation which means there is a normal LDLR from her mother (but you didn't check the molecular genetic diagnosis for her mother), the PCSK9i should be work. Therefore, other question is that based on current LDL-C results, is it necessary to use PCSK9i in the treatment?

3. For the figure of family tree, the author said molecular genetic diagnosis was only finished in the proband and her father. I suggest that suspected FH patients who have not undergone a genetic test need to place a question mark. In addition, place the highest LDL-C level under each people may be better to understand.

4. Is it possible to perform LDLR function tests on the patient's peripheral blood using flow cytometry? And do some bioinformatics analysis to predict LDLR function (including protein structure changes, etc.)? It is possible to develop treatment strategies tailored to patients based on the results of functional experiments.

Author Response

We want to thank for the comments along with interesting suggestions. We revised our paper in order to address point-by-point each comment.

Comments and Suggestions for Authors

The authors reported a novel large deletion of the LDLR gene 29 involving exons 4–18, identified by bioinformatic analysis of NGS data in an Italian family. Using the NGS to identify CNVs is very important, however, there are concerns listed below.

  1. There are some papers about identify the CNVs in NGS data (such as Genes (Basel). 2022;13(8):1424), Is there anything special about the methods or techniques in your article compared to other articles? what are the highlights of your article?

In agreement with the Reviewer, we introduced a new paragraph describing the methodological highlights of our work in the context of relevant literature (259-267 plus reference).

  1. For the proband, I'm interesting in her LDL-C levels. It seemed that PCSK9i didn't work for this patient. The first LDL-C level was 225mg/dl in 1994, but it still had 236mg/dl in 2022. Could you explain why used Ezetimibe plus Alirocumab can't reduce LDL-C levels? Might because the large deletion of the LDLR gene cause the dysfunction of LDLR? But it is a heterozygous variation which means there is a normal LDLR from her mother (but you didn't check the molecular genetic diagnosis for her mother), the PCSK9i should be work. Therefore, other question is that based on current LDL-C results, is it necessary to use PCSK9i in the treatment?

We thank the reviewer for the interesting observation.

Certainly, the presence of a large deletion in the LDLR gene, including most of the functional protein domains, could probably result in no LDLR production (null allele), thus contributing to a massive reduction in LDLR-mediated endocytosis of LDL protein.

This could explain the lower efficacy of PCSK9i observed in this patient, even though she is an FH carrier

In addition, we have no data on the off-therapy lipid panel. We can therefore speculate, considering that hypercholesterolemia worsens with age, that the patient currently has higher off-therapy LDL-C levels than in 1994. It is therefore likely that PCSK9 inhibition has minimal hypolipidemic effect.

In addition, the patient is intolerant to statins, so the important synergistic effect of statin therapy with ezetimibe and PCSK9i is missing.

  1. For the figure of family tree, the author said molecular genetic diagnosis was only finished in the proband and her father. I suggest that suspected FH patients who have not undergone a genetic test need to place a question mark. In addition, place the highest LDL-C level under each people may be better to understand.

In agreement with the Reviewer, we added the question marks in the new version of the Figure 1 changing the caption. Unfortunately, we cannot provide the LDL-C levels of the other members of the family due to the lack of this type of detailed information from the patient.

  1. Is it possible to perform LDLR function tests on the patient's peripheral blood using flow cytometry? And do some bioinformatics analysis to predict LDLR function (including protein structure changes, etc.)? It is possible to develop treatment strategies tailored to patients based on the results of functional experiments.

We thank the reviewer for the interesting suggestion. Unfortunately, is not possible to add experimental information using flow cytometry or biochemical methods. Moreover, the bioinformatics tools that generally can be used for the evaluation of a mutation effect on protein structure/function are not suitable for CNV analysis. However, we can provide some speculations about the effect of the new rearrangement on the LDLR protein (232-235 plus references).

Reviewer 2 Report

I have carefully reviewed your manuscript titled "Identification and Molecular Characterization of A Novel Large-Scale Variant (Exons 4_18 Loss) in the LDLR Gene as A Cause of Familial Hypercholesterolaemia in An Italian Family", and found it to be interesting and well-executed. However, there are still a few points that need to be addressed before it can be considered for publication.

Data Availability: In the manuscript, you mentioned that the data are available upon request. While this is acceptable, I strongly encourage you to deposit the sequencing-related data, such as the NGS data, into a publicly accessible database like NCBI. Please consider uploading your data to a public platform and provide the corresponding accession numbers or links in the manuscript. This would enhance the transparency and reproducibility of your research.

Clarification of Figure 2b: In the manuscript, it is unclear whether Figure 2b represents the MLPA results of the proband or the proband's father. If it represents the proband's MLPA results, the description in the manuscript stating that the deletion was simultaneously detected in the father's sample is contradictory. Please clarify this point and, if applicable, provide the MLPA results of the proband's father in the manuscript.

Case Report Content: In a case report, it is important to provide a comprehensive description of the patient's clinical history, relevant diagnostic findings, treatment interventions, and follow-up outcomes. Please ensure that all necessary information is included in the manuscript to provide a complete and informative case report.

Overall, your manuscript presents valuable findings in the field of familial hypercholesterolemia. Addressing the above-mentioned points will further enhance the quality and impact of your work. I look forward to receiving the revised version of your manuscript.

Author Response

We want to thank for the comments along with interesting suggestions. We revised our paper in order to address point-by-point each comment.

I have carefully reviewed your manuscript titled "Identification and Molecular Characterization of A Novel Large-Scale Variant (Exons 4_18 Loss) in the LDLR Gene as A Cause of Familial Hypercholesterolaemia in An Italian Family", and found it to be interesting and well-executed. However, there are still a few points that need to be addressed before it can be considered for publication.

Data Availability: In the manuscript, you mentioned that the data are available upon request. While this is acceptable, I strongly encourage you to deposit the sequencing-related data, such as the NGS data, into a publicly accessible database like NCBI. Please consider uploading your data to a public platform and provide the corresponding accession numbers or links in the manuscript. This would enhance the transparency and reproducibility of your research.

We thank the reviewer for the interesting suggestion. We submitted the information about the new LDLR variant to NCBI database ClinVar and we waiting for the submission ID.

Clarification of Figure 2b: In the manuscript, it is unclear whether Figure 2b represents the MLPA results of the proband or the proband's father. If it represents the proband's MLPA results, the description in the manuscript stating that the deletion was simultaneously detected in the father's sample is contradictory. Please clarify this point and, if applicable, provide the MLPA results of the proband's father in the manuscript.

We thank the reviewer for the observation. In the “Results” section (paragraph 3.1) we cited the Figure 2b with reference to the proband’s evaluation and we stated that “Successively, the proband’s father, screened by LDLR MLPA analysis, resulted carrier of the deletion”. However, we provide a new version of the manuscript with a more clear statement about the MLPA result of the proband’s father (183).

Case Report Content: In a case report, it is important to provide a comprehensive description of the patient's clinical history, relevant diagnostic findings, treatment interventions, and follow-up outcomes. Please ensure that all necessary information is included in the manuscript to provide a complete and informative case report.

In agreement with the Reviewer, we think that all the relevant clinical and laboratory data should be reported in the paper. According to the clinical professional members of the authors team, we described all the available information.

Overall, your manuscript presents valuable findings in the field of familial hypercholesterolemia. Addressing the above-mentioned points will further enhance the quality and impact of your work. I look forward to receiving the revised version of your manuscript.

Round 2

Reviewer 1 Report

The present form is better. However, there still have one thing should be emphasized.

1. I think the aim of genetic detection is to identify and treat disease, it is also very important to apply it to clinical practice after discovering genetic variations. In this case, the authors screened the proband's family, which is good. However, did you give some suggestions for those who may have the same muations to have lipid-lowing therapies right now? On the other hand,  PCSK9i is obviously ineffective for this case. It is expensive and why we still use it? Can you recommend other treatment methods? For example, the new ANGPT3 inhibitor, Evinacumab may be more suitable for her. 

Author Response

We thank the reviewer for the observation. We evaluated the patient's father, who is being treated by another Hospital where they live in southern Italy. He is on statin, ezetimibe, and PCSK9i therapy and has LDL-C values essentially on target.

We think that one of the main problems of our patient’s therapy, is her statin intolerance. She had previously experienced muscle pain during simvastatin, atorvastatin, and lovastatin therapy, although not correlated with a CPK elevation. So, as explained in the text, at the last visit we attempted a rechallenge with rosuvastatin 5 mg that is still ongoing. We think that the synergistic effect between rosuvastatin even if low dose, with alirocumab, may give us some results. Unfortunately, other drugs targeting ANGPT3 are not yet available in Italy, such as Evinacumab or Vupanorsen, that certainly could be more effective.

While waiting for these new therapies, in case the patient will not tolerate even rosuvastatin, we could give her bempedoic acid (a drug recently made reimbursable by the Italian national health care system) although this drug has a reduction effect on LDL cholesterol close to 20 percent.

We better detailed this aspect in the new version of the manuscript (section Case presentation and family history)